# Effect of the Sterile Insect Technique and Augmentative Parasitoid Releases in a Fruit Fly Suppression Program in Mango-Producing Areas of Southeast Mexico

**DOI:** 10.3390/insects14090719

**Published:** 2023-08-22

**Authors:** Jorge Cancino, Pablo Montoya, Fredy Orlando Gálvez, Cesar Gálvez, Pablo Liedo

**Affiliations:** 1Programa Moscas de la Fruta SENASICA-SADER, Camino a los Cacaoatales S/N, Metapa de Domínguez 30860, Chiapas, Mexico; 2Instituto de Biociencias, Universidad Autónoma de Chiapas, Boulevard Akichino S/N, Tapachula 30798, Chiapas, Mexico; pablo.montoya@unach.mx; 3Comité Estatal de Sanidad Vegetal de Chiapas (CESAVECHIS) SENASICA-SADER 4a Calle Oriente, entre 1a y 3a Sur, Tapachula 30830, Chiapas, Mexico; fredyoc24@gmail.com; 4Departamento de Control Biológico (CNRCB), SADER-SENASICA-DGSV, Km. 1.5 Carretera Tecomán Estación FFCC, Tecomán 28110, Colima, Mexico; cesarbiotecnologia@gmail.com; 5El Colegio de la Frontera Sur (ECOSUR), Carretera Antiguo Aeropuerto Km. 2.5, Tapachula 30700, Chiapas, Mexico; pliedo@ecosur.mx

**Keywords:** fruit fly management, area-wide control, fruit fly parasitism, FTD

## Abstract

**Simple Summary:**

The integrated use of the Sterile Insect Technique (SIT) and Augmentative Biological Control (ABC) in the management of fruit fly pest populations has been theoretically proposed to generate a synergistic effect. In a control program against the Mexican fruit fly, *Anastrepha ludens* (Loew), the combined application of the SIT and ABC was evaluated in mango orchards. The release of parasitoids alone increased parasitism percentages from 0.59 to 19.38%, and the application of the SIT reduced the FTD index (Flies per Trap per Day) by 30%. The concurrent application of both techniques resulted in about 98% suppression in the fly population. These results justify the use of both techniques in fruit fly area-wide pest management programs.

**Abstract:**

The Sterile Insect Technique (SIT), by means of sterile male releases of *Anastrepha ludens* (Loew), coupled with Augmentative Biological Control (ABC), by releasing the parasitoid *Diachasmimorpha longicaudata* (Ashmead), was evaluated in a commercial mango production area for one year. The obtained results were compared with mean fruit fly population values from two previous years without the combined use of both techniques. The treatments were: SIT + ABC, SIT, ABC, and Control, and each treatment was established in blocks of 5000 Ha separated by distances of 5–10 km. The evaluations were carried out through fruit sampling to assess percent parasitism and trapping of adult flies to obtain Flies per Trap per Day (FTD) values. The mean percentage of parasitism increased from 0.59% in the control treatment to 19.38% in the block with ABC. The FTD values decreased from ~0.129 and ~0.012 in the control block to 0.0021 in the block with SIT and ABC, representing a 98% suppression. The difference between the two periods in the control block was not significant. We conclude that the integration of both techniques resulted in an additive suppression of the pest population, supporting the use of both control techniques in an area-wide pest management context.

## 1. Introduction

The sterile insect technique (SIT) has been used successfully in different pest control programs [1]. The first and most successful case has been the eradication of the cattle screwworm *Cochliomyia hominivorax* (Coquerel) (Diptera: Calliphoridae) in North and Central America [1,2]. The SIT has been widely applied in the control of fruit flies (Diptera: Tephritidae) throughout diverse parts of the world with different objectives and results [3,4,5].

The use of the SIT for fruit fly suppression, containment, or eradication of populations [5] has been possible throughout the establishment of mass rearing facilities for different fruit fly species [6,7]. The production of high quality and competitive sterile males, is the most important requirement to successfully apply the SIT in the field, making possible the required sterility induction into the pest population of the target species. The existence of these rearing facilities makes also possible the rearing of some fruit fly parasitoid species to be used in augmentative biological control (ABC) programs [8]. The augmentative releases of parasitoids have also shown effectiveness in suppressing pest populations [9,10,11], mainly attacking fruit fly larvae inside the fruits. Both techniques share important advantages and common approaches, such as their high specificity and low environmental impact [12,13,14,15]. In addition, it is considered that their effect can be complementary when used together.

The SIT consists of the massive releases of sterile insects, which, when mating with the wild ones, induce sterility in the wild population, reducing its growth rate. In the ABC, adult parasitoids are massively released to parasitize their hosts. In tephritid fruit flies, the two techniques control two different stages of flies’ populations. Theoretical models have proposed the combined use of both techniques, since it is argued that a synergistic effect may be obtained, achieving thus a greater suppression of the pest population [16,17]. Because each technique is used against a specific developmental stage of fruit flies, theoretical models have proposed that combined use of both techniques may render a synergistic effect in the suppression of the pest population [16,17].

In the case of fruit flies, the theoretical models proposed by Barclay [18] and Knipling [19] concluded that the combined effect of these techniques would be more effective than their separate effects. One of the challenges of this proposed strategy has been to empirically evaluate its impact under area-wide field conditions, since this type of evaluation requires enough biological material (i.e., sterile insects, parasitoids), release operation facilities, and field areas with homogeneous conditions for the evaluation of each treatment.

In the coast of the state of Chiapas, located in Mexico Southeast, the local mango producers’ association carries out an area-wide integrated pest management (AW-IPM) program to suppress *Anastrepha ludens* (Loew), the main pest of mango Ataulfo cultivar [20]. The AW-IPM program includes a trapping system to monitor adult populations, fruit sampling to monitor parasitism percentages in pest populations, the use of bait stations as mass-trapping in commercial orchards, destruction of fallen infested fruits, ground bait sprays focused on reservoir sites, and the application of hot-water postharvest treatment for fruits destined for the export market [20,21,22]. The control activities based on the SIT and ABC currently are maintained throughout the year on a total surface of 18,000 ha, with the purpose of establishing a biological barrier between refuge zone and the zone where most commercial mango orchards are located, to minimize the movement of flies during the mango season [20].

The mass-rearing facility, Moscafrut (SENASICA-SADER), in Metapa, Chiapas, is located in this region, which implicates an important advantage for the evaluation of sterile flies and parasitoid releases. This facility has the capacity to produce up to 300 million *A. ludens* sterile flies of a bisexual strain, 50 million of a genetic sexing strain (Tapachula-7) [23] and 50 million *D. longicaudata* parasitoids, per week [23,24]. In addition, there are irradiation facilities and the infrastructure for emerging the sterile flies and parasitoids for ground and aerial release [25,26]. These conditions afforded us the opportunity to carry out an open field evaluation of the application, alone and in combination, of the augmentative biological control method (ABC) and the sterile insect technique (SIT), to suppress *A. ludens* populations.

Our objective here was to compare the effect of ABC plus SIT with the effect of each control method alone upon open field conditions.

Here, we report the results obtained in the 2014 season releasing sterile *A. ludens* Tapachula-7 (genetic sexing strain) males and *D. longicaudata* parasitoids, both alone and in combination, by analyzing data on parasitism and adult fly trapping. Furthermore, these data were compared with results obtained in the two previous years from the same areas and with the control without releases.

## 2. Materials and Methods

### 2.1. Study Areas

The evaluation was carried out in blocks of approximately 5000 ha separated by distances between 5 and 10 km. The blocks were distributed throughout the mango-growing area of the Soconusco Region, Chiapas, which mainly includes the municipalities of Huehuetán, Mazatán, Tapachula, and Tuxtla Chico. The working area was basically established in the coastal plain of the Soconusco Region, which has an Am climate type according to Köpen and Geiger, with a mean temperature of 24.8 °C and a mean annual rainfall of 3843 mm. Sterile flies and parasitoids have been released weekly in the region since 2013. The evaluations of *Anastrepha* fly population and parasitism levels were carried out in 2014 in four different blocks, where a specific treatment was applied during this period. Figure 1 shows a map with the geographic distributions of the four blocks. The satellite image gives an idea of the landscape ecology and vegetation cover. The characteristics of each block are described below:

#### 2.1.1. Raymundo Enriquez (ABC + SIT Treatment)

Area with abundant and diverse secondary vegetation corresponding to high evergreen forest but altered by the presence of different small crops of cocoa, flowers, corn, papaya, and a high diversity of fruit trees that host *Anastrepha* spp., such as chicozapote, mamey, orange, guava, and creole mango. Control activities are carried out to suppress *Anastrepha* pest populations in this “marginal zone”, minimizing thus the invasion of neighboring commercial mango production areas [27]. During the evaluation period, *D. longicaudata* parasitoids and sterile *A. ludens* flies (Tapachula-7) were released weekly (ABC + SIT). The monitoring of adult flies was carried out by the weekly inspection of 16 traps.

#### 2.1.2. Mazatán (SIT Treatment)

The mango cultivar Ataulfo is dominant in this area, which is surrounded by extensive banana and soybean crop areas. There are also small areas of wild vegetation classified as middle tropical rainforest combined with small crops and the presence of fruits that are alternate hosts of *Anastrepha* such as creole mango and guava, which are found in lower numbers and at a lower density than in the previous area. Control activities in this area are more regular and are applied for the specific purpose of reducing *Anastrepha* spp. fly populations. The release of sterile *A. ludens* (Tapachula-7) males was applied in this area (SIT). Twenty-three traps were placed for the sampling of adult flies.

#### 2.1.3. Tuxtla Chico (ABC Treatment)

This block is characterized by the presence of small areas with the production of cocoa, annual crops such as corn, sesame, sorghum, etc., combined with areas of other crops (tomato, squash, etc.) and fruit trees, some of which are alternate hosts of *Anastrepha* flies. There are also small mango commercial orchards. The work program for the control of fruit flies places great emphasis on using biological control and exclusively applies the release of parasitoids. Therefore, this block was selected for the ABC treatment. In this area, 20 traps were placed to monitor pest fly populations.

#### 2.1.4. Huehuetán (Control Treatment)

In this block, Ataulfo mango orchards are dominant, but there is also a high density of other fruit trees that are hosts of *Anastrepha* flies. However, there was no regular control program for fruit fly populations at an area-wide level, and thus control activities were conducted only in specific areas, mainly by applying bait sprays. This area was used as a control without the release of parasitoids or sterile flies, and 22 traps were placed to sample fly populations.

In the whole region, since 2012 a trapping network was maintained, which included the four described blocks. This allowed us to have a reference of the Flies per Trap per Day (FTD) values as an indicator of fly population levels. Fruit sampling for parasitism evaluation also was constant in this period. However, fruits were randomly collected throughout the entire region; there were no specific data for these four blocks. In the period of 2012–2013, the FTD data corresponded to fly populations without the application of any treatment in any of the blocks. The data obtained in 2014 corresponded to the results of the application of the described treatments in each block. The number of traps per block varied (16–22 traps) depending on the accessibility or importance of the zone for obtaining the indicators of fly population levels. Fruits presumed infested with *Anastrepha* larvae were sampled weekly to determine the parasitism percentage.

### 2.2. Biological Material

*Diachasmimorpha longicaudata* parasitoids and sterile *A. ludens* male flies of the genetic sexing strain Tapachula-7 were released into the experimental blocks of the corresponding treatments. The insects were provided by the Moscafrut facility located in Metapa de Dominguez, Chiapas, Mexico. The parasitoids were produced using irradiated (45 Gy) 9-day-old *A. ludens* larvae to prevent the emergence of non-parasitized hosts. In the case of sterile *A. ludens* adults, 13-day-old and pupae marked with dye-glo powder were subjected to 80 Gy radiation using a GB-127 gamma radiator (Nordion International Inc. model, Ottawa, ON, Canada) with a Cobalt-60 source [23].

### 2.3. Packing and Release of Insects

#### 2.3.1. Parasitoids

Ten thousand 14-day-old pupae (one day before emergence) of the parasitoid *D. longicaudata* were placed in an “Arturito” type container (36.3 cm diameter × 30.2 cm height 20 L cylindrical plastic container) with three circular windows of 30 cm in diameter covered with 1.5 mm mesh [28]. About 80% of adult emergence occurred in the first four days at 26 °C. The adults were kept at 20 °C for another two days to facilitate mating and reduce stress and were fed with honey. The ground release of adults was carried out on the seventh day, for which the containers with parasitoids were moved to the different release points established based on the presence of fruit infested with *Anastrepha* larvae.

#### 2.3.2. Flies

Thirty thousand *A. ludens* (Tapachula-7) pupae were placed inside a plastic container (50 cm × 15 cm × 5 cm) with longitudinal perforations through which emerged flies were able to exit. The container was placed inside a cage with an aluminum frame measuring 80 cm × 70 cm × 10 cm covered with mesh. Each cage is stackable and an accumulation of 16 levels forms a “Mexico” Tower. The flies were fed with a mixture of hydrolyzed yeast and sugar (1:24) placed in a tray (60 cm × 6 cm × 2.5 cm) and hydrated with a water-saturated sponge covered with a cloth bag [29,30]. A 110 cm × 40 cm folded piece of plastic cardboard was included in each cage to increase the resting area for the flies and reduce stress. The towers were kept at 23 ± 1 °C for 5 days, at the end of which sterile males reached sexual maturity. For the release, the towers were placed in a cold room at 2–4 °C for 60 min [25]. Once the flies were torpid, they were placed in a release box [30], which was suitable for transportation in a Cessna 206 airplane used for the aerial release of sterile males. Both flies and parasitoids were released every week throughout the year 2014.

### 2.4. Distribution of Insect Releases

The sterile flies were aerial released to get a homogeneous distribution throughout the 5000 ha of each block. In the case of parasitoids, the ground releases were carried out early in the morning using a pickup and were focused to reservoir zones with high density of *Anastrepha* spp. host trees with larval infestation. An average of 3.5 million parasitoids were released per week with a density of ~1500 parasitoids per ha. The flies were released at a density of ~850 males/ha [21].

### 2.5. Indicators of Parasitism and Fly Population Levels

To measure the effect of parasitoid releases, samples of likely infested fruits were collected, while traps were used to measure the effect of sterile male releases on the adult population. Both activities were carried out preferably in the central part of each block to minimize the border effect.

#### 2.5.1. Parasitism

Fruit sampling included various fruits infested with 2nd and 3rd instar larvae of *Anastrepha* spp. Based on fruit size, samples of 1–2 kg were collected for small fruits (hog plums, guavas, etc.), of 3–4 kg for medium size fruits (mangoes, chicozopote, oranges, etc.), and of 5 kg or more for large fruits (large mangoes, mamey, grapefruit, etc.), as in Montoya et al. [27]. The fruits were dissected in the laboratory and the number of *Anastrepha* larvae was counted. The larvae were placed in plastic containers (7.5 cm in diameter × 4.3 cm in height) with vermiculite and kept for 15 days at 26 °C and 60–80% RH until adult emergence. The emerged flies [31] and parasitoids [32] were identified based on morphological characters using taxonomic keys. Emergence percentage was calculated as: % emergence = (No. of parasitoids + No. of flies × 100)/No. of larvae. Parasitism percentage was calculated as: % parasitism = (No. of parasitoids × 100)/(No. of parasitoids + No. of flies) [9].

#### 2.5.2. Flies per Trap per Day (FTD)

The FTD index was obtained through the trapping net, based on the number of flies captured. Traps were placed in continuous routes in each block. Multilure traps (Multilure^®^ Better World, Fresno, CA, USA) baited with Biolure^®^ Suterra LLC, Inc., Bend, OR, USA (a mixture of ammonium acetate and putrescine) and propylene glycol was used to retain the attracted flies. The traps were checked weekly and captured adult *A. ludens* flies were removed with entomological forceps and identified based on morphological characters using taxonomic keys [31]. Wild *A. ludens* flies and released (marked) sterile flies were differentiated by the head dye glo marking previously described. Only the number of wild flies was used to calculate the FTD index as follows: FTD = Captured wild flies/(Number of traps) × (exposure days).

### 2.6. Data Analysis

The mean larvae/fruit values and parasitism percentages obtained from the fruit sampling in each experimental block were analyzed by means of a non-parametric Kruskal–Wallis test. Parasitism means were compared with a paired test using the Wilcoxon method. The FTD data were analyzed using a bifactorial design with period (2012–13 vs. 2014) and block as factors. A GLM with a logit link function was used for this analysis, and the mean FTD values of each block were compared between periods with Student’s *t*-test. All analyses were performed in JMP version 16 (SAS Institute Inc., Cary, NC, USA).

## 3. Results

### 3.1. Parasitism

The releases of *D. longicaudata* in the ABC and ABC + SIT treatments showed notable parasitism levels in the different fruits sampled (Table 1). In medium-sized fruits, such as guavas and creole mangoes, mean parasitism percentage was higher than 20%, but it was higher in small fruits (e.g., hog plums up to 58%). The lowest values were observed in large fruits such as sour orange, grapefruit, and mamey (up to 10%). In blocks where parasitoids were not released (SIT and Control), mean parasitism ranged between 0.59 and 2.53%. Mean parasitism percentage was significantly higher in the blocks where parasitoids were released (d.f. = 3, χ^2^ = 9.28, *p* = 0.02). *Diachasmimorpha longicaudata* was present in all experimental blocks and was the dominant parasitoid species. In the case of fruits with higher parasitoid diversity, *D. longicaudata* shared parasitism (~70%) with other native species, with a mean parasitism percentage of less than 10%. There was no relationship between parasitism level and fruit infestation level (larvae/fruit), which did not decrease in the areas where sterile males were released (d.f. = 3, χ^2^ = 6.67, *p* = 0.08).

The fly species that emerged from the infested fruits could be grouped according to the taxonomic family of the fruits: the Rutaceae (citrus) were infested by *A. ludens*, the Zapotaceae by *A. serpentina* (Wiedd.), the Anacardiaceae (mango and hog plum) by *A. obliqua,* and the Myrtaceae (guava) by *A. striata* (Loew). The parasitoid *D. longicaudata* showed higher parasitism percentages in introduced exotic fruits, while parasitism was generally shared with native parasitoids species in native fruits (caimito, guava, mamey, and plums).

### 3.2. Flies per Trap per Day (FTD)

*Anastrepha ludens* populations were the second most dominant according to the FTD index derived from the trapping system. More than 90% of the flies trapped corresponded to *A. obliqua*, a species that rarely infests the mango cultivar Ataulfo (Aluja et al.) [33]. Figure 2 shows the mean FTD values of *A. ludens*, where the lowest values were observed when sterile flies were released, followed by the block with releases of sterile flies and parasitoids, and the block where only parasitoids were released had higher values. However, the highest values were obtained in the control block, where no releases were made. The analysis of the FTD data showed significant differences between blocks (χi^2^ = 50.06, d.f. = 3, *p* < 0.0001), no significant differences between periods (χi^2^ = 0.09, d.f. = 1, *p* = 0.07), and no significant interaction between the two factors (χi^2^ = 0.25, d.f. = 3, *p* = 0.96).

When comparing the years 2011 and 2012 (without releases) with the year 2014 (with insect releases) different results were observed in each block. The mean FTD value decreased in the areas with the release of parasitoids and sterile flies. The FTD index decreased by 82.5% (from 0.012 to 0.0021) in 2014 in the block where both parasitoids and sterile flies were released, where the difference in FTD between the two years without releases and the year with releases was significant (t = −1.83, d.f. = 30, *p* = 0.03). In the area where only sterile flies were released, there was a decrease of about 30% (from 0.0033 to 0.0023), although it was not significant (t = −1.60, d.f. = 44, *p* = 0.27). In the block where only parasitoids were released, there was a reduction of 18% (from 0.028 to 0.023) but the difference was not significant (t = −0.50, d.f. = 38, *p* = 0.30).

In the control block, the population of *A. ludens* increased by up to 7% with respect to the previous years (from 0.129 to 0.139), although without statistical differences (t = −0.69, d.f. = 42, *p* = 0.24) (Figure 3). The FTD values of *A. ludens* obtained in 2014 showed a clear decrease compared to those of the years 2012 and 2013. The decrease in the FTD index in the block with parasitoid and sterile fly releases was of more than 80%, in the block with sterile fly releases, it was a little higher than 30%, and in the block with only parasitoid releases, it was of 17.85%. In contrast, FTD levels increased in the control block (Figure 3).

## 4. Discussion

The data obtained in 2014, when parasitoids and sterile flies were released weekly, showed a stronger suppressive effect when both techniques were concurrently applied, where there was an 82.5% decrease in FTD compared to when the techniques were applied separately. The suppressive effect on the FTD index was higher with the release of sterile flies, while parasitism percentage increased significantly with the exclusive release of parasitoids.

Regarding the use of the SIT, different studies have shown results of pest population suppression levels that are considered sufficient to achieve eradication in a region [5,21]. Although there are few studies providing experimental field data to support this assumption, the SIT has been considered a key strategy for maintaining a low prevalence or areas free of fruit flies in different regions [34,35,36,37]. In the case of *A. ludens*, there are data that support the effect of sterile fly releases causing a significant reduction in wild populations [38], especially with the use of the genetic sexing strain Tap-7 [39,40,41].

In the case of *D. longicaudata* releases, different studies have reported successful fly population suppression [11,42,43,44]. Montoya et al. [10] demonstrated that augmentative releases of *D. longicaudata* achieved effective suppression of *Anastrepha* spp. populations in mango orchards in the coastal zone of Chiapas, Mexico. In addition, Montoya et al. [45] showed that releases of *D. longicaudata* increase parasitism levels in “marginal areas” with high fruit infestation rates by *Anastrepha* larvae in the same geographical region. The same study also showed that *D. longicaudata* prefers free hosts, apparently to avoid competition with native parasitoids, which can be explained under the concept of taking advantage of “free spaces” [46,47]. In the present work, the parasitism percentages in *Anastrepha* larvae with native parasitoid species remained at values that are normally reported independently of the release of *D. longicaudata* in different regions [10,48,49]. *Diachasmimorpha longicaudata* can thus be considered a species that has a complementary effect on the parasitism of *Anastrepha* larvae, reinforcing the coexistence with native species of the region.

Different studies have proposed that ABC and SIT can be complementary and synergistic techniques, which implies that the simultaneous use of both techniques would result in a greater reduction in pest populations [17,18,19]. A study by Wong et al. [50] showed that populations of *Ceratitis capitata* (Wiedemann) that infest coffee plants in Maui, Hawaii were suppressed by releasing sterile flies and the parasitoid *Diachasmimorpha tryoni* (Cameron). Similar results were reported by Sivinski et al. [51] in areas with coffee plants infested by *C. capitata* in Guatemala.

Our results show that releases of parasitoid and sterile fly suppressed the fly population significantly when compared with the application of any of the two techniques separately. This finding supports the assumption that the concurrent use of these techniques has a desirable complementary effect, suggesting that this strategy could be used advantageously in efforts to suppress, contain, or eradicate populations of fruit fly pests. This proposal is supported by data obtained from an experiment in field cage conditions [52]. In addition, the joint use of these techniques has two important advantages: the specificity of both techniques and their complementary effect, since parasitoids attack the larval stage of the flies, causing lower emergence rates of wild adults, and thus favoring the proportion of sterile males to wild males [52].

Among the multiple factors that can affect each technique, there are some of higher priority that require greater attention for a better use and management. In the case of parasitism as an indicator of parasitoid effectiveness, it is influenced by the host-fruit larva–parasitoid relationship. As has been mentioned in previous studies, a first challenge is the size or shape of the fruit [53,54,55], since parasitism percentage did not reach 10% in large fruits, such as citrus or mamey, and the opposite occurred in small fruits, such as some guavas or hog plums. Another important factor is the response of parasitoids to the emission of semiochemicals from the feeding activity of the larvae in different fruits [56,57]. The parasitoid *D. longicaudata* has shown a wide adaptation to search for and develop in larvae of different *Anastrepha* species, mainly those infesting exotic fruits such as mango and citrus. However, the response of this parasitoid has not always been efficient in more specific conditions such as in native fruits of Neotropical origin [49]. One example is the caimito (*C. cainito*), which is a native fruit of small size highly infested by *A. serpentina*, where *D. longicaudata* has not been reported to reach high parasitism levels [58,59].

In the case of fly trapping, the FTD index is a more specific and direct measure of pest suppression. Although Multilure traps can catch different species of fruit flies, the obtained values are more direct and influenced by less factors than parasitism indicators. This allows having a more accurate indicator of fly populations. Our results indicate that the decrease in the fly population measured by the FTD index was caused by the application of the SIT but not by ABC, since the suppression by the parasitoids was not significant when applied alone, but it was when combined with the SIT.

Open field evaluation of area-wide pest management strategies represents difficulties from perspective of experimental design and the logistic of experimentation. The two main problems in the present work were: (1) the lack of homogeneity in the management of fly populations between the experimental blocks, and (2) the lack of consistency in the ecological conditions of these areas. The main problem associated with these issues in each working area was the specific agricultural production interests related to mango production [10,22]. Despite these limitations, our findings were consistent when we made comparisons, both in space (areas) and time (years). We observed a greater suppression of fruit fly pest populations with the concurrent use of both techniques, which was even greater than the results obtained with the separate application of each technique. However, this cannot be considered as a proper synergistic effect because the parameters obtained were not sufficiently robust to support this assumption [60,61]. The most important conclusion of this work is that our results reinforce the proposal of combining the SIT and ABC as complementary methods within the integrated management of fruit flies under an area-wide context.

## Figures and Tables

**Figure 1 insects-14-00719-f001:**
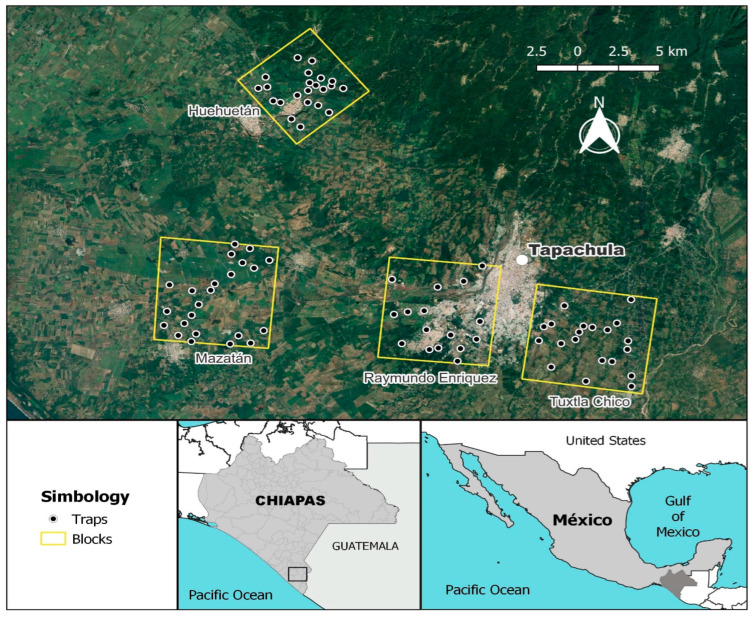
Geographical distribution with satellite image of the four working blocks to evaluate the effect of augmentative releases of parasitoids and sterile insects in the Soconusco Region, Chiapas.

**Figure 2 insects-14-00719-f002:**
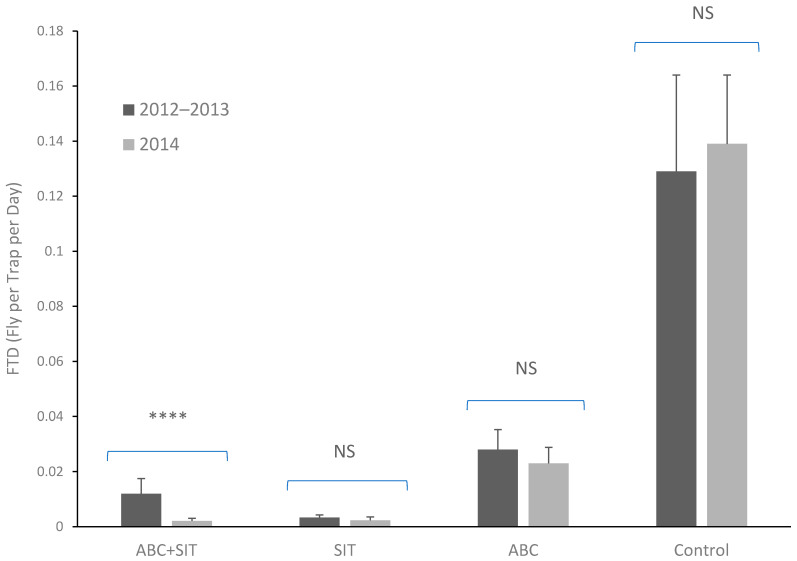
Flies per Trap per Day (FTD) values of *A. ludens* in the blocks with different treatments. Lines above bars indicate the S.E., **** = statistical difference and NS = non-statistical difference between periods in each block, Student’s *t*-test.

**Figure 3 insects-14-00719-f003:**
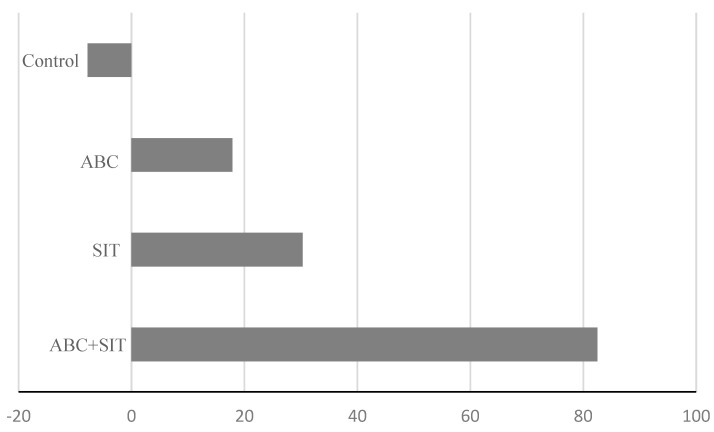
Percentage of decrease in the FTD index of *A. ludens* populations in the year with sterile fly and parasitoid releases (2014) with respect to the two previous years (2012−2013).

**Table 1 insects-14-00719-t001:** Results of fruit sampling during 2014 in the different experimental blocks with ABC and SIT and the respective controls. Fly species and infestation rate per fruit and parasitism percentage per species are shown.

Treatment	Host Fruits				Flies				Parasitoids				
						Emergence				Emergence			Parasitism *
	Species	NL ^+^		L/F ^+^	*Sp*F ^+^	NF^+^	%	TF ^+^	*Sp*P ^+^	NP ^+^	%	TP ^+^	
**ABC + SIT**	Caimito, *Chrysophyllum cainito* (L.)	127		10.71	*A. serpentina*	735	100	735	*D. longicaudata*	67	72.04	93	11.23
									*O. anastrephae*	26	27.96		
	Guava, *Psidium guajava* (L.)	2519		1	*A. str* *iata*	1506	98.69	1526	*D. longicaudata*	424	99.76	425	21.78
					*A. obliqua*	20	1.31		*A. pelleranoi*	1	0.24		
	Mamey, *Mammea americana* (L.)	970		9.59	*A. serpentina*	5348	100	5348	*D. longicaudata*	340	98.55	345	6.06
									*D. crawfordi*	5	1.45		
	Mango var. Ataulfo, *Mangifera indica* L.	232		0.18	*A. ludens*	14	73.68	19	*D. longicaudata*	6	100	6	24
					*A. obliqua*	5	26.32						
	Creole mango, *Mangifera indica* L.	4207		1.3	*A. ludens*	437	20.26	2157	*D. longicaudata*	1479	99.60	1485	40.77
					*A. obliqua*	1720	79.74		*D. areolatus*	3	0.20		
									*U. anastrephae*	3	0.20		
	Sour orange, *Citrus aurantium* L.	640		2.1	*A. ludens*	775	100	775	*D. longicaudata*	78	100	78	9.14
	Hog plums, *Spondias mombin* L.	2018		2.15	*A. obliqua*	1378	100	1378	*D. longicaudata*	1767	90.94	1943	58.51
									*D. areolatus*	66	3.40		
									*O. anastrephae*	7	0.36		
									*U. anastrephae*	103	5.30		
Mean				**3.86 ± 1.64 a**			100						**24.49 ± 7.20 a**
**SIT**	Grapefruit, *Citrus paradisi* Mcfad	160		0.46	*A. ludens*	50	100	50	*D. longicaudata*	4	100	4	7.41
	Guava, *Psidium guajava* (L.)	685		1.33	*A. striata*	765	100	765	*D. longicaudata*	10	100	10	1.29
	Creole mango, *Mangifera indica* L.	162		0.01	*A. obliqua*	1	100	1					
	Sour orange, *Citrus aurantium* L.	760		0.95	*A. ludens*	477	100	477	*D. longicaudata*	7	100	7	1.45
Mean				**0.68 ± 0.28 a**									**2.53 ± 1.65 b**
**ABC**	Grapefruit, *Citrus paradisi* Mcfad	80		0.74	*A. ludens*	39	100	39					
	Guava, *Psidium guajava* (L.)	33		1.45	*A. striata*	17	100	17	*D. longicaudata*	4	80	5	22.73
									*D. areolatus*	1	20		
	Mamey, *Mammea americana* (L.)	4		21	*A.serpentina*	67	100	67	*D. longicaudata*	7	100	7	9.46
	Mango var. Ataulfo, *Mangifera indica* L.	744		3.32	*A. ludens*	680	99.85	681	*D. longicaudata*	417	100	417	37.98
					*A. obliqua*	1	0.15						
**ABC**	Creole mango, *Mangifera indica* L.	1141	2.77		*A. ludens*	423	54.94	770	*D. longicaudata*	395	89.16	443	36.52
					*A. obliqua*	347	45.06		*D. areolatus*	48	10.84		
	Sour orange, *Citrus aurantium* L.	26	4.69		*A. ludens*	88	100	88	*D. longicaudata*	9	90	10	10.2
									*D. crawfordi*	1	10		
	Hog plums, *Spondias mombin* L.	176	3.15		*A. obliqua*	427	100	427	*D. longicaudata*	65	65.66	99	18.82
									*D. areolatus*	32	32.32		
									*A. pelleranoi*	2	2.02		
Mean			**5.3 ± 2.66 a**										**19.38 ± 5.36 a**
**Control**	Creole mango, *Mangifera indica* L.	1371	1.39		*A. ludens*	46	4.23	1088	*D. longicaudata*	13	100	13	1.18
					*A. obliqua*	1042	95.77						
	Bitter orange, *Citrus aurantium* L.	20	1.25		*A. ludens*	10	100	10					
Mean			**1.32 ± 0.07 a**										**0.59 ± 0.59 b**

^+^ Symbology: NL, Number of larvae; L/F, Larvae/Fruit; S*p*F, Species of flies; NF, Number of flies emerged; TF, Total number of flies; S*p*P, Species of parasitoids; NP, Number of parasitoids emerged; TP, Total number of parasitoids. * Means ± S.E. followed by different letters in infestation rate and parasitism percentage indicate significant differences between blocks. Non-parametric Kruskal–Wallis test.

## Data Availability

The raw data used for this manuscript were up-loaded to Zenodo under doi: 10.5281/zenodo.8004112.

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
