# Peer review of "Effect of the Sterile Insect Technique and Augmentative Parasitoid Releases in a Fruit Fly Suppression Program in Mango-Producing Areas of Southeast Mexico"

_insects, 2023, doi:10.3390/insects14090719_

Round 1
Reviewer 1 Report
1. The title refers to a specific program “Fruit Fly Suppression Program in Mango-Producing Areas” about which the introduction does not provide much information.
2. In the Introduction, the purpose of the work is correctly presented, but the Introduction is very concise. For a better understanding of the content of the article, a brief explanation of the SIT technique and the ABC method would be advisable.
3. Materials and methods: a figure (map) having the location of sampling and more detailed information on the atmospheric conditions in the study area would be desirable. There is no information on how often fruit’s samples were taken for analysis in the case of parasites. Were FTD analyzes performed weekly throughout the year? So, if the number of weeks were at least 48 the number of samples was the same?
4. The comparison of 2011 and 2012 (without releases) with 2014 (with releases of insects) is discussed, but there is no data anywhere that this comparison results from. Table 1 contains only data from 2014. The results include an interesting compilation of data on both the pest species and the fruit on which they were found. In what units is the FTD given, number of individuals, average number per trap? The caption of the figure 1. needs to be completed.

Author Response
Response to Reviewers, Insects-2465357, R1
Reviewer 1
- The title refers to a specific program “Fruit Fly Suppression Program in Mango-Producing Areas” about which the introduction does not provide much information.
R: The introduction was modified (lines 48-103), including a description of the area-wide fruit fly program in the mango producing region and references about it. In lines 75-86 we did some changes focusing the activities of the regional program into which were carried out the evaluations here reported.
- In the Introduction, the purpose of the work is correctly presented, but the Introduction is very concise. For a better understanding of the content of the article, a brief explanation of the SIT technique and the ABC method would be advisable.
R: We added information expanding the characteristics of both techniques. (Lines 48-63).
- Materials and methods: a figure (map) having the location of sampling and more detailed information on the atmospheric conditions in the study area would be desirable.
R: A new figure (Fig. 1) with a map showing the location of the study area and satellite image showing the landscape is included now. We also add information about climate variables, see, lines 110-112, page 3.
There is no information on how often fruit’s samples were taken for analysis in the case of parasites.
R: Fruit samples were collected every week in 2014 (Lines 161-162).
Were FTD analyses performed weekly throughout the year? So, if the number of weeks were at least 48 the number of samples was the same?
R: The FTD index was obtained weekly for 52 weeks per year through the 3 years. Fruit samples to determine parasitism were only collected in 2014 and there were weeks without fruit. We explain why we did not consider fruit sampling in 2012-2013 (Lines 154-156).
- The comparison of 2011 and 2012 (without releases) with 2014 (with releases of insects) is discussed, but there is no data anywhere that this comparison results from. Table 1 contains only data from 2014. The results include an interesting compilation of data on both the pest species and the fruit on which they were found. In what units is the FTD given, number of individuals, average number per trap? The caption of the figure 1. needs to be completed.
R: Table 1 refers only to percent parasitism, so there is only data from 2014, as explained above. Results from trapping were shown in Fig. 1 (now Fig. 2). The FTD values for the two previous years was the total average, so we compared the mean of the two previous years, with the mean of 2014. The annual FTD mean was the weekly average of the traps in that each particular block.
Reviewer 2 Report
Objectives were not clearly identified and presented.
Introduction does not provide any meaningful summary of the work.
There is not sequential presentation of material and methods.
All methods are fragmentary and do not portray the clear picture of what they have done.
They should have done this kind of study in the glasshouse conditions prior to the field study.
Each treatment was done in different farms and years that is not comparable with the data which was not carried out at the same time and conditions.
Overall, the manuscript does not qualify for the publication from my viewpoint.
Must be improved lot
Author Response
Response to Reviewers, Insects-2465357, R1
Reviewer 2
- Objectives were not clearly identified and presented.
R: Our objective was to compare the effect of ABC plus SIT with the effect of each control method alone, under open field conditions. We rewrite this in the introduction (Lines 97-98).
- Introduction does not provide any meaningful summary of the work.
R: We modified the introduction trying to provide a clearer background and justification for our research question (Lines 17-24, 26-39, 59-61).
- There is not sequential presentation of material and methods. All methods are fragmentary and do not portray the clear picture of what they have done.
R: We modified the order and description of our methods (see lines 201-206, 152-156), hoping they are now clearer and more understandable.
- They should have done this kind of study in the glasshouse conditions prior to the field study.
R: As we cited and discussed, a previous study was performed under field cage conditions (Montoya et al 2023, reference No. 52). Our aim here was to scale up the evaluation going from field cage to open field conditions, as close to real conditions as possible.
- Each treatment was done in different farms and years that is not comparable with the data which was not carried out at the same time and conditions.
R: Our experimental design considered variations in time and space. We had the treatments and the control in the same year, so the temporary conditions were similar in the four blocks. Then, we compared each treatment with a control in the same area, so the spatial conditions for each treatment were similar.
We are aware of the difficulties in conducting research in open field conditions and over large areas. It is almost impossible to have homogeneous conditions. But this type of research is necessary, since it includes factors that are not present in laboratory or greenhouse conditions, they are closer to reality and fortunately we had the conditions to do it.
The clear differences between the treatments, the consistency in the spatial and temporal comparisons, and the statistical analysis of the data make our results robust and reliable.
- Overall, the manuscript does not qualify for the publication from my viewpoint.
R: We disagree and regret the reviewer’s opinion. We hope once our arguments are known and the modifications made on the manuscript, will change his/her opinion.
Reviewer 3 Report
The introduction part is informative giving basic informations about Sterile insect technique and biological control.
In the materials and methods, a map showing locations of study areas.
Results, discussion and conclusions are well explained
Author Response
Response to Reviewers, Insects-2465357.
Reviewer 3
- The introduction part is informative giving basic informations about Sterile insect technique and biological control.
- We are including a figure with a map to show the location of the study area and a satellite image to show the landscape.
- Results, discussion and conclusions are well explained.
- Thanks
Round 2
Reviewer 2 Report
Not convinced with revised version of manuscript because the whole experimental process is very vague and intricate. The most important thing is that methods and materials are not sequential, data analysis is not clear, not showing any credential analysis undertaken. Therefore, the manuscript does not deserve to be published.